# Development of a Hallmark Pathway-Related Gene Signature Associated with Immune Response for Lower Grade Gliomas

**DOI:** 10.3390/ijms231911971

**Published:** 2022-10-09

**Authors:** Guichuan Lai, Xiaoni Zhong, Hui Liu, Jielian Deng, Kangjie Li, Biao Xie

**Affiliations:** Department of Epidemiology and Health Statistics, School of Public Health, Chongqing Medical University, Yixue Road, Chongqing 400016, China

**Keywords:** lower grade gliomas, signaling pathways, immune response, allograft rejection, biomarkers, prognosis, immunotherapy, chemotherapeutic drugs

## Abstract

Although some biomarkers have been used to predict prognosis of lower-grade gliomas (LGGs), a pathway-related signature associated with immune response has not been developed. A key signaling pathway was determined according to the lowest adjusted p value among 50 hallmark pathways. The least absolute shrinkage and selection operator (LASSO) and stepwise multivariate Cox analyses were performed to construct a pathway-related gene signature. Somatic mutation, drug sensitivity and prediction of immunotherapy analyses were conducted to reveal the value of this signature in targeted therapies. In this study, an allograft rejection (AR) pathway was considered as a crucial signaling pathway, and we constructed an AR-related five-gene signature, which can independently predict the prognosis of LGGs. High-AR LGG patients had higher tumor mutation burden (TMB), Immunophenscore (IPS), IMmuno-PREdictive Score (IMPRES), T cell-inflamed gene expression profile (GEP) score and MHC I association immunoscore (MIAS) than low-AR patients. Most importantly, our signature can be validated in four immunotherapy cohorts. Furthermore, IC50 values of the six classic chemotherapeutic drugs were significantly elevated in the low-AR group compared with the high-AR group. This signature might be regarded as an underlying biomarker in predicting prognosis for LGGs, possibly providing more therapeutic strategies for future clinical research.

## 1. Introduction

Gliomas are common brain tumors with a poor prognosis [1]. Gliomas were traditionally classified into four grades, in which grades I–II were referred to as low-grade glioma in the past [2]. However, grades II–III were later defined as lower-grade gliomas (LGGs), including oligodendrocytomas and astrocytomas according to the WHO classification [3]. Although a better prognosis appears in LGG patients compared with glioblastoma (GBM) patients, most LGG patients eventually develop to high-grade gliomas, leading to a resistance to various treatments, such as chemotherapy and radiotherapy [4]. There are still insufficient effects for risk stratification in LGGs, although several clinical biomarkers have been used in the case of the management of LGGs prognosis [5]. Therefore, a novel predictive signature is needed to improve the survival of LGGs.

Most patients show a limited sensitivity to these approaches due to tumor heterogeneity while several treatments have already brought therapeutic efficacy for LGG patients [6]. Given the crucial role of the tumor immune microenvironment in tumor heterogeneity, immunotherapy has been considered as a novel promising therapy against various solid tumors [7]. Recently, more and more studies have focused on the impact of immune checkpoint inhibitors (ICIs) on the prognosis of LGGs. For example, an early clinical study demonstrated that the majority of LGG patients displayed specific immune responses to one or more IMA950 antigens [8]. Mair et al. found that LGG patients receiving bevacizumab had lower soluble PD-L1 levels than those at baseline [9]. However, a limited anti-tumor immunity or immune response becomes the major obstacle for the utilization of immunotherapy [10,11,12]. How to distinguish between “cold” and “hot” tumors is always a challenge for many clinical researchers. It is clear that immune infiltration provides more opportunities to improve the ability of anti-tumor immune response [13,14]. Meanwhile, a clear recognition on the classification of the tumor immune microenvironment is beneficial for identifying more patients who respond to immunotherapy, thereby improving the efficacy of immunotherapy [15].

The association of certain pathway activities with cancer progression has been demonstrated in some studies. For example, the STAT3 signaling pathway is regarded as a potential target in cancer immunotherapy [16]. The STING pathway develops anti-tumor activity through tumor-originated DNA reacting and T cell activating [17]. Although some useful pathway-related signatures were utilized to predict the prognosis of cancer patients, few studies integrated these signaling pathways with the immune microenvironment [18,19,20]. Therefore, it is necessary to construct a pathway-related signature to predict the LGG patients’ prognosis and to improve their responses to therapies.

In this study, we developed a predictive pathway-related biomarker associated with immune response considering the importance of immune subtypes on immunotherapy. Single sample gene set enrichment (ssGSEA), “ESTIMATE”, and gene set variation algorithms (GSVA) were used to identify “cold” and “hot” tumors, followed by determining the most valuable pathway. Some pathway-related genes were collected from the gene set enrichment analysis (GSEA) database. The least absolute shrinkage and selection operator (LASSO)-Cox model was used to construct a novel gene signature based on these pathway-related genes. Five datasets from CGGA, ArrayExpress, and Rembrandt databases were regarded as the external validation sets. Immune infiltration analysis, somatic mutation analysis, and some novel predictive ICI-related algorithms were applied to evaluate our signature in predicting the efficacy of ICIs. Drug sensitivity analysis on six classic chemotherapeutic drugs was used to select the patients who were more sensitive to these chemotherapeutic agents. The main purpose of our study was to accurately predict the prognosis of LGGs and exert the potential target to therapies through our pathway-related signature based on bioinformatics.

## 2. Results

### 2.1. The Characteristic of Immune Subtypes

Two immune subtypes were identified based on 28 immune cells of 509 TCGA-LGG patients according to the lowest PAC value (Figure 1A–C). Patients with cluster 1 had higher immune, stromal and estimate scores but lower tumor purity than those with cluster 2 (Figure 1D,E). The results of GSVA-KEGG showed that higher immune-related pathway scores were enriched in the patients with cluster 1 (Figure 1F). Furthermore, GSEA-KEGG also reflected that the biological activities in patients with cluster 1 were associated with some immune-related functions, such as B cell receptor signaling and T cell receptor signaling pathways (Figure 1G,H).

### 2.2. The Definition of “Cold” and “Hot” Tumors

Considering that patients with cluster 1 presented more immune-related traits based on the results above, we defined tumors in cluster 1 as the “hot” tumors and tumors in cluster 2 as the “cold” tumors. We detected that 47.08% of the patients who received radiotherapy were hot-tumor patients, while only 27.65% of the patients who did not receive radiotherapy were hot-tumor patients (χ2 = 15.756, *p* < 0.001) (Figure 2A). The ratio of patients with hot tumors in the radiotherapy group was almost twice as high as in non-radiotherapy group. However, these patients with “hot” tumors had a worse overall survival (OS) than those with “cold” tumors (Figure 2B).

### 2.3. Allograft Rejection (AR) was a Key Signaling Pathway for LGGs

GSVA analysis on 50 hallmark pathways revealed a key signaling pathway between “hot” tumors and “cold” tumors. The AR pathway was chosen for the key signaling pathway because of the lowest adjusted p value (Figure 2C). In addition, scores of this signaling pathway were negatively associated with a favorable OS (Figure 2D).

### 2.4. Construction and Validation of an AR-Related Gene Signature

We discovered 79 differentially expressed AR-related genes, including 74 upregulated genes and 5 downregulated genes. Then, 60 prognostic genes were selected for LASSO, ridge and elastic net regression analyses. The results showed that the highest concordance appeared in the LASSO regression model, but the lowest concordance was found in the ridge regression model (Appendix A). Therefore, we chose the LASSO as the main selective method. After performing the LASSO analysis, BRCA1, ABI1, CAPG, FLNA, STAT1 and EIF3D were screened as the candidate genes. Finally, the stepwise multivariate Cox model was used to construct an AR-related five-gene signature. AR score = 0.78507 × BRCA1 − 0.51643 × ABI1 + 0.29859 × CAPG + 0.20447 × FLNA − 0.52151 × EIF3D. Survival analysis showed that LGG patients with high AR scores from TCGA, CGGA, Rembrandt, and ArrayExpress databases tended to have a poor OS (Figure 3A,D–H). The AUC of 1-, 3-, and 5-year OS in TCGA was 0.87, 0.88, and 0.78, respectively (Figure 3B). The AUC of 1-, 3-, and 5-year OS in other datasets was shown in Appendix A. AR score and survival distribution maps of LGG patients were exploited to further assess the discriminatory power in TCGA (Figure 3C).

### 2.5. Verification of Gene Expression at the Protein Level

The staining of BRCA1, ABI1, and EIF3D was medium in tumor tissue but low in normal tissue. The staining of CAPG was low in tumor tissue but undetectable in normal tissue. The staining of FLNA was high in tumor tissue but low in normal tissue (Appendix A). All these results indicated that higher expressions of these five AR-related genes might be in tumor tissue than in corresponding normal tissue, which was consistent with the results from GEPIA (Appendix A).

### 2.6. Establishing an Individualized Nomogram

Age, grade, and AR score were associated with prognosis (*p* < 0.05) in univariate Cox regression analysis (Figure 4A), and these prognostic factors were integrated to construct a predictive indicator displayed via nomogram analysis (Figure 4B). As a result, AR score accompanied by the highest predictive power independently predicted the OS of LGGs. A goodness of fit between the prediction and observation can be observed by the calibration plot (Figure 4C).

### 2.7. Tumor Immune Microenvironment and AR-Related Gene Signature

Higher infiltrations of CD8^+^ T cells, M1 and M2 macrophages, and resting mast cells but lower proportions of activated mast cells and eosinophils were found in the high-AR group than low-AR group (Figure 5A). Additionally, LGG patients in the high-AR group had a higher level of CD8^+^ T cells via “MCP-counter” (Figure 5B) and “TIMER” algorithms (Figure 5D). LGGs expressed higher M1 and M2 macrophages by “quanTIseq” (Figure 5C) and “xCell” algorithms (Figure 5E). Furthermore, low-AR LGG patients showed higher eosinophils through the “Xcell” algorithm (Figure 5E). All these results showed that CD8^+^ T cells, M1 and M2 macrophages, and eosinophils were differentially distributed between high- and low-AR groups, indicating a potential association of our signature with the tumor immune microenvironment.

### 2.8. The Correlation between AR-Related Gene Signature and TMB

The analysis of somatic mutation revealed that alteration frequency of IDH1 was the highest mutated gene among the top 10 driver genes (Figure 6A). There was an increased TMB in the high-AR group (Figure 6B). Most importantly, the AR-related gene signature significantly showed survival differences in both the high-TMB group and low-TMB group, presenting that the AR-related gene signature was a potential predictor with independence of TMB (Figure 6C).

### 2.9. AR-Related Gene Signature, IDH1 Mutation and TMB

A higher proportion of patients with IDH1 mutation was found in the low-AR group while a higher proportion of patients with wild-type IDH1 appeared in the high-AR group in TCGA (χ2 = 9.309, *p* < 0.01) (Figure 7A), CGGA-325 (χ2 = 25.282, *p* < 0.001) (Figure 7B), CGGA-693 (χ2 = 59.680, *p* < 0.001) (Figure 7C) and E-MTAB-3892 (χ2 = 4.333, *p* = 0.037) (Figure 7D). Furthermore, LGG patients with mutated IDH1 had a lower AR score than those with wild-type IDH1 (Figure 7E–H)). Next, we found that patients with wild-type IDH1 in the high-AR group expressed higher TMB than those in the low-AR group (Figure 8A). However, no statistical difference of TMB could be observed between these two groups of patients with IDH1 mutation (Figure 8B).

### 2.10. The Relationship between AR-Related Gene Signature and Histology

In order to explore the relationship between AR-related signature and histology, we investigated whether the proportion of histology was differentially distributed between high- and low-AR groups. The results showed that a higher proportion of astrocytoma patients but a lower ratio of oligodendroglioma patients was gathered in the high-AR group compared to those in the low-AR group in TCGA (χ2 = 38.836, *p* < 0.001) (Figure 9A), CGGA-325 (χ2 = 4.677, *p* = 0.031) (Figure 9B) and CGGA-693 (χ2 = 4.499, *p* = 0.034) (Figure 9C). Most importantly, an obvious distinction of AR scores was detected in different histological subtypes, showing that astrocytoma patients experienced higher AR scores than the oligoastrocytoma and oligodendroglioma patients (Figure 9D–F).

### 2.11. The Significance of AR-Related Gene Signature in Targeted Therapies

CD274, CTLA4, HAVCR2, PDCD1LG2, and PDCD1 were the upregulated immune checkpoints (ICs) in the high-AR group (Figure 10A). With an increase in AR score, expressions of these ICs were additionally increased (Figure 10D–H). In addition, the biological functions of these differentially expressed genes (DEGs) between high- and low-AR groups were related to some immune-related activities and pathways, such as T cell activation, MHC protein complex, cytokine activity, and Th1 and Th2 cell differentiation (Figure 10B,C). Striking GEP and MIAS score differences were observed between high- and low-AR groups, indicating that high-AR LGG patients were more likely to respond to PD-1 blockade therapy (Figure 10I,J). Similarly, there was more IMPRES and IPS noted in the high-AR group, suggesting a higher probability of response to immunotherapy in these patients with high AR scores (Figure 10K,L). These results above revealed that LGG patients may show a partial response to immunotherapy according to our risk stratification. In the subsequent analysis, our signature was an effective prognostic biomarker in both metastatic urothelial patients who received anti-PD-L1 treatment in the IMvigor210 cohort and metastatic melanoma patients receiving anti-PD-1 therapy in Liu’s cohort (Figure 11A,D). A significantly elevated AR score appeared in patients responding to anti-PD-L1 in the IMvigor210 cohort, and the overall response rate was higher in the high-AR group than in the low-AR group (Figure 11B,C). Patients responding to anti-PD-1 in Liu’s cohort had higher AR scores than those non-responders, and a higher overall response rate was observed in the high-AR group (Figure 11E,F). Decreased AR scores emerged in GBM and chronic lymphocytic leukemia patients responding to ICIs (Figure 11G,I). The AUC was 1 and 0.76 in GSE79671 and GSE148476, respectively, manifesting a high value of our signature in predicting their overall responses (Figure 11H,J). We observed that estimated IC50 values of the six classic chemotherapeutic drugs were significantly elevated in the low-AR group compared with the high-AR group, indicating that high-AR LGG patients were more sensitive to these drugs (Figure 12A–F).

## 3. Discussion

With an increasing number of studies aimed at the tumor immune microenvironment, new approaches targeted at immunotherapy were gradually explored. Immune cells exert the role of a double-edged sword in involving killing or promoting the development of tumor cells by immune-activated or immune-suppressive microenvironments. Therefore, seeking the immune subtypes is a reasonable way to face this challenge. In this study, we constructed a novel signature associated with AR signaling pathway that distinguished between the “cold” and “hot” tumors.

In the present study, two immune-related subtypes were determined based on 28 immune cells through consensus clustering analysis. We found that there were more immune, stromal and estimate but fewer tumor purity scores in patients with cluster 1. Furthermore, LGG patients in cluster 1 were subjected to more immune-related pathways, such as the B cell receptor signaling pathway, allograft rejection and T cell receptor signaling pathway, showing higher pathway scores. Most notably, the result of GSEA enrichment analysis showed that PD-L1 expression and the PD-1 checkpoint pathway in cancer and Th17 cell differential pathways were enriched in the cluster 1 group. In contrast, patients with cluster 2 expressed the contrary signaling pathway with less immune-related function. Therefore, we tried to define the “hot” and “cold” tumors according to the results above.

We ultimately identified a key signaling pathway called AR from 50 hallmark pathways between “hot” and “cold” tumors. Then, the highest enrichment score of the AR pathway was found in the patients with cluster 1 (hot tumors), indicating that the AR pathway might be more relevant to the development of lower grade glioma (LGG) patients with hot tumors compared with other hallmark pathways. This AR process was associated with the role of alloantigen on innate immunity [21]. It was found that an innate immune mechanism emerged in the AR and proinflammatory cytokines activated by innate immune-stimulated T cell expansion during this pathway process [22]. Meanwhile, Ashwin et al. observed that AR was an unregulated pathway in the immune response, and Subhayan et al. detected that this pathway was related to the B cell receptor signaling pathway [23,24]. The association between AR and gliomas has been reported. The AR pathway is the most important pathway and was involved in the process of immune response of GBM in a previous study [25]. Most importantly, Zhou et al. found that AR was highly associated with EGFR amplification, showing inconsistent activity statuses across the LGG population, and it was dysregulated in more than 50% of LGG samples [26].

Although various machine learning methods were used to select appropriate variables, how to determine an optimal method still remains a problem. In this study, we compared LASSO with other machine methods (elastic net and ridge regression) to further emphasize the importance of the LASSO model. Eventually, we constructed an AR-related signature, which comprised five AR-related genes (BRCA1, ABI1, CAPG, FLNA and EIF3D). BRCA1 promoted the GBM cell growth and negatively decreased the survival of gliomas [27,28]. It was noted that BRCA1 was a risk prognostic factor for LGGs but not for GBM patients [29]. The loss of ABI1 promoted an aggressive development in tumor cells, leading to shorter survival in GBM [30]. CAPG participating in immune escape was identified as a poor prognostic gene for LGG patients, of which its higher expression was found in tumor tissues versus that in normal tissues [31,32,33]. FLNA was a novel driver gene of tumor metastasis in GBM, and EIF3D was a protective gene for gliomas, of which high expression of EIF3D was related to an improved OS [34,35]. A nomogram analysis displayed that our signature accounted for higher predictive value than other clinical features. Higher expressions of these five genes at the protein level were found in the LGG tissues versus those in corresponding normal tissues through the HPA database, suggesting that the five AR-related genes were possible diagnostic biomarkers of LGGs.

In this work, we found that a higher proportion of astrocytoma patients appeared in the high-AR group while a higher proportion of oligodendroglioma patients was gathered in the low-AR group in TCGA, CGGA-325 and CGGA-693. These results indicated that different histological compositions were observed between high- and low-AR groups. Furthermore, we also found that astrocytoma patients expressed the highest AR scores while oligodendroglioma patients showed the lowest AR scores, suggesting that histological subtypes were closely related to our AR-related signature. In previous studies, astrocytoma patients demonstrated to experience the worst prognosis compared to other histology patients [36,37,38]. In addition, Xu et al. and Yang et al. found that the highest-risk scores were enriched in astrocytoma patients, but the lowest-risk scores were observed in oligodendroglioma patients [39,40]. These findings were similar to ours.

In terms of the tumor immune microenvironment, we found higher infiltrating levels of CD8^+^ T cells and M1 and M2 macrophages but a low proportion of eosinophils in the high-AR group, which was similar to the result of Zhang et al. [41]. Macrophages were mainly divided into M1 and M2 macrophages. M1 macrophages represented the proinflammatory subtype, activating the immune response, while M2 macrophages constructed an immunosuppressive microenvironment through anti-inflammatory activity [42]. M1 macrophages as drug carriers accounted for tumor inhibition in gliomas [43]. However, higher M1 macrophages correlated with poor OS in grade II and III gliomas [44]. Likewise, Lin et al. found that high M1 macrophages were positively related to high risk scores using an autophagy-related signature associated with the tumor microenvironment in LGGs [45]. In addition, LGGs with higher immune scores had a worse prognosis and higher infiltration of M2 macrophages [46]. CD8^+^ T cells were essential for T cell-mediated tumor control and could release more predictive ICI-related biomarkers, such as PD-1 and PD-L1 [47]. Yang et al. constructed an ICI score based on 22 immune cells and ascertained that higher eosinophils were in the high-ICI LGG patients who had a worse prognosis [48]. Furthermore, we used several immune infiltrating methods to ensure the stability of our results on the basis of these results above. Therefore, our signature might provide more clues for future studies on the tumor immune microenvironment.

Although more and more cancer patients were receiving immunotherapy, an optimal level was limited to reach. TMB was a potential biomarker for predicting the efficacy of ICIs through displaying higher neoantigens [49]. Patients with high TMB level were more likely to respond to ICIs because of a high likelihood of recognition by neoantigen-reactive T cells [50]. However, how to determine an optimal cutoff value of TMB remained a big challenge for ICIs. It was found that a high TMB was negatively associated with the OS because of high infiltration of CD8^+^ T cells and macrophages in LGGs [51]. In this study, a higher TMB level was found in the high-AR group, indicating a potential value of our signature in the ICIs. Furthermore, patients with IDH1 mutation expressed lower AR scores in this study, indicating that patients with IDH1 mutation had a lower risk of developing a poor prognosis compared to those with wild-type IDH1, further confirming previous conclusions [52,53]. It was noted that the IDH1 mutation could suppress CD8^+^ T cell accumulation in patients with gliomas [54]. Furthermore, we also found that a higher proportion of patients with IDH1 mutation appeared in the low-AR group than the high-AR group, suggesting that fewer CD8^+^ T cells were observed in the low-AR group. This may be one of the reasons why a higher proportion of CD8^+^ T cells was enriched in the high-AR group. In previous studies, IDH1 mutation was associated with low TMB [51,55]. Therefore, we compared the level of TMB between high- and low-AR groups in patients with IDH1 mutation or wild-type IDH1 in order to investigate whether the IDH1 mutation impacted the relationship between AR scores and TMB. We found that an elevated level of TMB was noted in the high-AR patients with wild-type IDH1, but no statistical difference of TMB between the high- and low-AR groups was observed in patients with the IDH1 mutation, indicating that the IDH1 mutation may not be an indicator in evaluating the role of AR-related signatures in immunotherapy. These DEGs between high- and low-AR groups were involved in some immune-related pathways or biological activities. Some important ICs, such as CD274, CTLA4 and PDCD1, showed a strong positive correlation with our signature. PD-L1 expression promoted an active immune microenvironment, and patients with PD-L1-positive status were more likely to respond to immunotherapy than those with PD-L1-negative status in various solid tumors [56]. A higher level of CTLA4 was found in these melanoma patients responding to ipilimumab treatment than in non-responders [57]. PD-L2 could independently exert predictive power in the progression-free survival, and a longer survival was observed in the cancer patients with PD-L2-positive status than those with PD-L2-negative after receiving pembrolizumab therapy [58].

With more valuable and predictive immune-related genes being selected for developing a considerable scoring algorithm, some novel algorithms were utilized for predicting the efficacy of ICIs. Most importantly, the predictive power of these algorithms has been demonstrated through various real immunotherapy cohorts, especially for melanoma patients. In this study, we chose the IPS, GEP, IMPRES and MIAS algorithms to evaluate the value of our signature in predicting LGG patients’ response to ICIs. All these results showed an association of higher probability of ICI response with higher AR score, further emphasizing the potential role of our signature in immunotherapy. Furthermore, we used several solid tumors with real immunotherapy data to validate the predictive power considering these novel ICI-related biomarkers developed based on melanoma patients. Therefore, four different cancer patient cohorts receiving immunotherapy from GEO datasets and previous literature reports were obtained for predicting the patients’ prognosis and efficacy of immunotherapy. We found that these patients could be effectively stratified, and AR scores were differentially expressed between responders and non-responders. Additionally, the resistance of LGG patients’ response to chemotherapeutic drugs also limits the application of some drugs, thus decreasing the survival of LGGs. In the present research, we used this pathway-related gene signature to predict LGG patients’ response to six classic chemotherapeutic drugs. The results revealed that high-AR LGG patients might be suitable for receiving the six drugs. Limited response affected the application of cisplatin, although cisplatin was used for chemotherapy against glioma [59]. These LGGs with IDH mutation accompanied with high expression of PTPRN or RIM-BP2 were most sensitive to cisplatin [60]. Sunitinib was demonstrated to be a potential candidate to suppress neovascularization in GBM [61]. The PI3K inhibitor plays a key role in the pathogenesis and progression of glioma [62]. These findings may provide clues for future clinical trials.

## 4. Materials and Methods

### 4.1. Sample Collection and Data Processing

The mRNA-seq and clinical data of LGGs from TCGA database were downloaded from UCSC Xena (https://xenabrowser.net/) website (accessed on 1 May 2022). Gene annotation file “Homo_sapiens.GRCh38.105.chr.gtf.gz” was downloaded from Ensembl (http://Asia.ensembl.org/) database (accessed on 1 May 2022). The mRNA sequencing and clinical data of CGGA-325 and CGGA-693 datasets were obtained from CGGA (http://www.cgga.org.cn/) database (accessed on 1 May 2022). Gene expression profiles in TCGA and CGGA were measured by the transcript per million (TPM) estimation and log_2_-based transformation. The mRNA microarray and corresponding clinical data from Rembrandt, E-MTAB-2768 and E-MTAB-3892 datasets were obtained from CGGA (http://www.cgga.org.cn/download_other.jsp) (accessed on 1 May 2022). and ArrayExpress (https://www.ebi.ac.uk/arrayexpress/) databases (accessed on 1 May 2022). The robust multi-array average algorithm by “affy” package was used for background correction and normalization [63]. In this study, grade II and III glioma patients with complete survival information were regarded as enrolled subjects. A total of 1427 LGGs from these datasets were included into this study. In addition, TCGA-LGG dataset was used as training dataset, and other datasets were used to validate the predictive power. The flow chart and data characteristics are shown in Appendix A and Table 1, respectively.

### 4.2. Immune Subtype Analysis

To quantify the tumor immune microenvironment, we collected 28 gene sets characterizing 28 immune cell types and used ssGSEA algorithm to score these immune cells by “GSVA” package [64]. We performed consensus unsupervised clustering analysis to classify LGG patients into different immune subtypes based on these 28 immune cells using “ConsensusClusterPlus” package. We adopted the “pam” algorithm with “euclidean” as a measure of distance, including 80% item resampling and 1000 repetitions. The optimal k was determined by the proportion of ambiguous clustering (PAC) and appeared when the PAC attached the lowest value [65]. “ESTIMATE” is a common quantitative algorithm for tumor microenvironment, which was used to speculate the immune, stromal and estimate scores, and tumor purity [66]. GSVA-KEGG analysis was used to calculate GSVA scores of 186 signaling pathways based on 186 gene sets by “GSVA” package. We used “clusterProfiler” package to perform GSEA-KEGG analysis in order to identify the difference of biological pathways based on these DEGs among different immune subtypes [67]. The pathways with an adjusted *p* < 0.05 were selected as differentially expressed pathways. In this study, we defined the “cold” and “hot” tumors according to these results above.

### 4.3. The Determination of a Key Signaling Pathway

In this study, we downloaded 50 hallmark gene sets from GSEA (https://www.gsea-msigdb.org/) database (accessed on 3 May 2022) and used “GSVA” package to score these 50 signaling pathways. The “limma” package was used to compare the GSVA scores between “cold” and “hold” tumors, and the key signaling pathway was determined by the lowest adjusted *p* value [20,68].

### 4.4. Construction and Validation of a Signaling Pathway-Related Gene Signature

We collected 200 genes associated with our key signaling pathway from GSEA database. Firstly, differentially expressed analysis was conducted through GEPIA (http://gepia.cancer-pku.cn/) database (accessed on 4 May 2022), containing 518 LGG samples and 207 adjacent normal samples. DEGs were extracted by |log_2_FC| > 1 and *p* value < 0.05. Secondly, we used “survival” package to perform univariate Cox regression analysis and identify prognostic genes where *p* < 0.05 was considered statistically significant. Thirdly, these prognostic genes were employed in LASSO, ridge and elastic net regression analyses with 1000 repetitions after dividing full datasets into training and testing datasets with a 2:1 ratio by “glmnet” package. The concordance reflects the predictive power of models. The best model was determined by the highest concordance. Subsequently, those genes appearing in over 500 repetitions in the best model were ultimately included into multivariate Cox regression model. Finally, a signaling pathway-related score was calculated by the following formula: Score = ExpGene1 × CoeGene1 + ExpGene2 × CoeGene2 + ExpGene3 × CoeGene3 + …… + ExpGeneN × CoeGeneN, where “Exp” represents the expression level and “Coe” represents the regression coefficient. LGG patients were divided into high- and low-score groups according to the median value. We used “timeROC” package to perform time-independent receiver operating characteristics (ROC) analysis. In addition, this constructed signature was externally validated in CGGA-325, CGGA-693, Rembrandt, E-MTAB-2768 and E-MTAB-3892 datasets.

### 4.5. Immunohistochemical (IHC) Staining Analysis

IHC staining images of the pathway-related genes in LGG tissues and normal brain tissues were obtained from the Human Protein Atlas (HPA) (http://www.proteinatlas.org/) database (accessed on 10 May 2022). The expression level of the target protein was classified into four degrees of staining, including high, medium, low, and not detected.

### 4.6. Construction of the Nomogram

Clinical features and our signaling pathway-related signature were simultaneously executed into the univariate Cox model to investigate whether our signature was an independent predictive factor. Variables with *p* < 0.05 in univariate Cox regression model were mapped into the nomogram to construct a prognostic signature for 1-, 3-, and 5-year OS by “rms” package. Points represent the importance of variables on the prognosis of patients, with higher points reflecting higher prognostic significance. To test the stability of the nomogram, calibration curves were used to compare the actual OS with the predictive OS.

### 4.7. Immune Infiltration Analysis

The proportion of 22 immune cells was evaluated by “CIBERSORT” algorithm (https://cibersort.stanford.edu/, accessed on 12 May 2022), a machine learning approach for characterizing the tumor immune microenvironment. Leucocyte signature matrix 22 (LM22) consisting of 547 genes was obtained from a previous study [69]. We ran the CIBERSORT algorithm with 1000 permutations, and patients with *p* < 0.05 were selected into the infiltrating analysis. The “TIMER”, “MCP-counter”, “quanTIseq” and “xCell” algorithms were used to ensure the stability of tumor-infiltrating results [70,71,72,73].

### 4.8. Somatic Mutation Analysis

We used “maftools” package to perform somatic mutation analysis to acquire the TMB of each patient [74]. The waterfall diagram was visualized by the “maftools” package. The relationship between AR-related signature and IDH1 mutation was explored in TCGA, CGGA-325, CGGA-693 and E-MATB-3892 datasets.

### 4.9. Prediction of Targeted Therapies

The “edgeR” package was used to screen out the DEGs between high- and low-score groups [75]. Then, we used “clusterProfiler” package to determine the biological function via GO and KEGG pathway enrichment analyses. In this study, we chose these five ICs (CD274, CTLA-4, HAVCR2, PDCD1LG2 and PDCD1) as the targeted ICs. Furthermore, we used four available algorithms that demonstrated a high predictive value in melanoma patients receiving ICIs. Immunophenscore (IPS) was measured on a 0–10 scale based on 162 represented genes [76]. The IPS was changed by Z-scores determined by cell type, including stimulatory and inhibitory factors. The IMmuno-PREdictive Score (IMPRES), which ranges from 0–15 scale, was calculated based on 15 immune-related gene pairs [77]. Patients with higher IPS and IMPRES were more likely to respond to ICIs. T cell-inflamed gene expression profile (GEP) score was evaluated through 18 inflammatory genes [78]. The MHC I association immunoscore (MIAS) was assessed based on 100 immune-positive MHC I-associated signature genes [79]. A higher probability of response to PD-1 blockade therapy appeared in patients with higher GEP score and MIAS. Finally, we collected several real immunotherapy cohorts including four solid cancers to evaluate whether our signature was an applicable potential target to immunotherapy. These cohorts comprised the GSE79671 dataset, GSE148476 dataset, IMvigor210 cohort, and Liu’s cohort [80,81]. There were 16 GBM patients receiving bevacizumab treatment in GSE79671, 62 chronic lymphocytic leukemia patients receiving ICIs in GSE148476, 348 metastatic urothelial patients receiving anti-PD-L1 immunotherapy in IMvigor210 cohort and 119 patients with metastatic melanoma receiving anti-PD1 therapy in Liu’s cohort. To evaluate the established pathway-related gene signature in the prediction of classic chemotherapeutic drugs, such as Cisplatin, GDC0941, Cyclopamine, Sunitinib, AZD6482 and Bleomycin, we calculated the IC50 value of the six classic drugs for LGG patients by “pRRophetic” package based on the Genomics of Drug Sensitivity in Cancer database [82].

## 5. Conclusions

In this study, we identified a crucial signaling pathway and constructed an AR-related gene signature, which contributed to predicting the prognosis and targeted therapies for LGGs. We believe our signature may provide a new sight for future research and guide the therapeutic strategy.

## Figures and Tables

**Figure 1 ijms-23-11971-f001:**
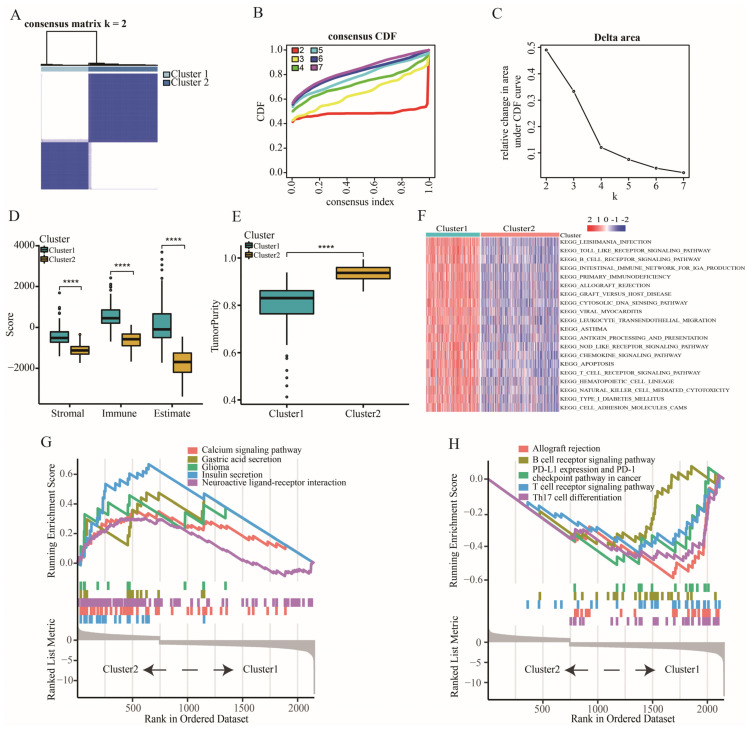
The classification and characteristic of immune subtypes. (**A**–**C**) Identification of two immune subtypes using consensus clustering analysis. (**D**) The expression of stromal, immune, and estimate scores between cluster 1 and cluster 2. (**E**) The difference of tumor purity between cluster 1 and cluster 2. (**F**) GSVA analysis of biological pathways between cluster 1 and cluster 2. (**G**) GSEA analysis showing the pathways enriched in cluster 2. (**H**) GSEA analysis showing the pathways enriched in cluster 1. Data in (**D**,**E**) were analyzed by Wilcoxon test; **** *p* < 0.0001.

**Figure 2 ijms-23-11971-f002:**
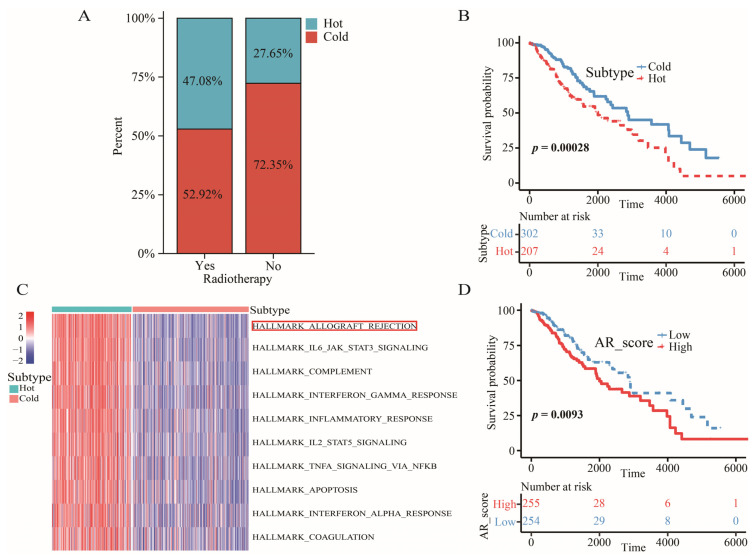
The characteristic of “hot” tumors, “cold” tumors, and AR signaling pathway. (**A**) The distribution of “hot” and “cold” tumors in patients with or without radiotherapy. (**B**) Kaplan–Meier survival analysis of OS between “hot” and “cold” tumors. (**C**) GSVA analysis of 50 hallmark pathways between cluster 1 and cluster 2. (**D**) Kaplan–Meier survival analysis of OS between high- and low-AR_score groups.

**Figure 3 ijms-23-11971-f003:**
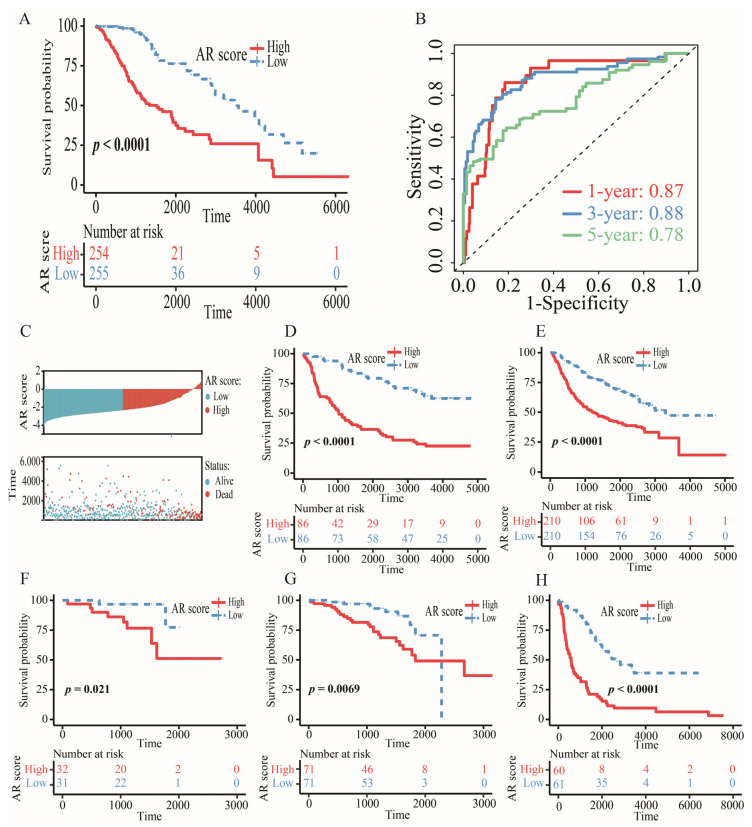
The prognostic value of AR-related gene signature. (**A**) Kaplan–Meier survival analysis of OS between high- and low-AR groups in TCGA. (**B**) The AUC of 1-, 3-, and 5-year OS in TCGA. (**C**) The distribution of AR score and survival status in TCGA. (**D**–**H**) Kaplan–Meier survival analysis of OS between high- and low-AR groups in CGGA-325, CGGA-693, E-MTAB-2768, E-MTAB-3892 and Rembrandt, respectively.

**Figure 4 ijms-23-11971-f004:**
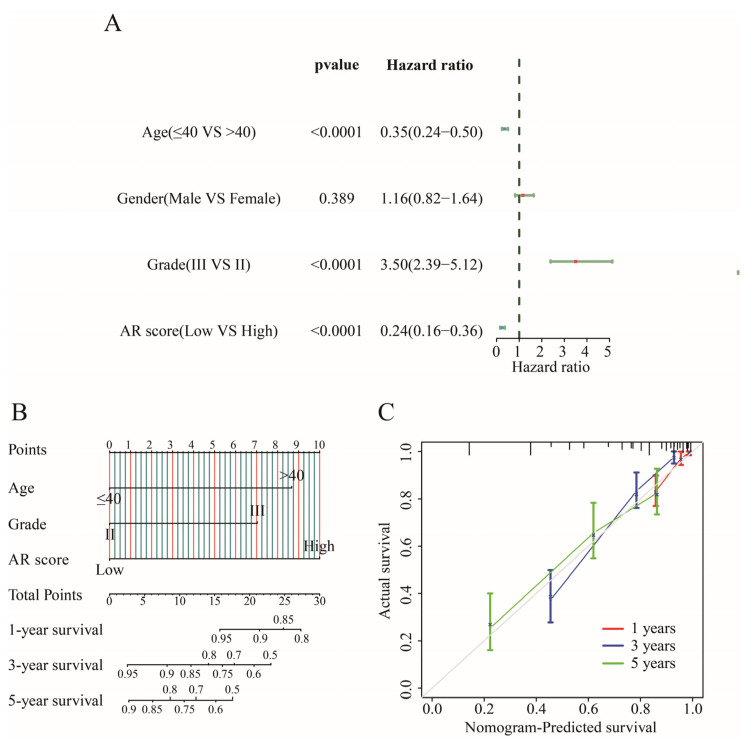
Construction of a nomogram. (**A**) The forest plot of univariate Cox regression analysis. (**B**) Nomogram for predicting the 1-, 3-, and 5-year OS of LGG patients. (**C**) Calibration curves of the nomogram for predicting of 1-, 3-, 5-year OS in patients with LGG.

**Figure 5 ijms-23-11971-f005:**
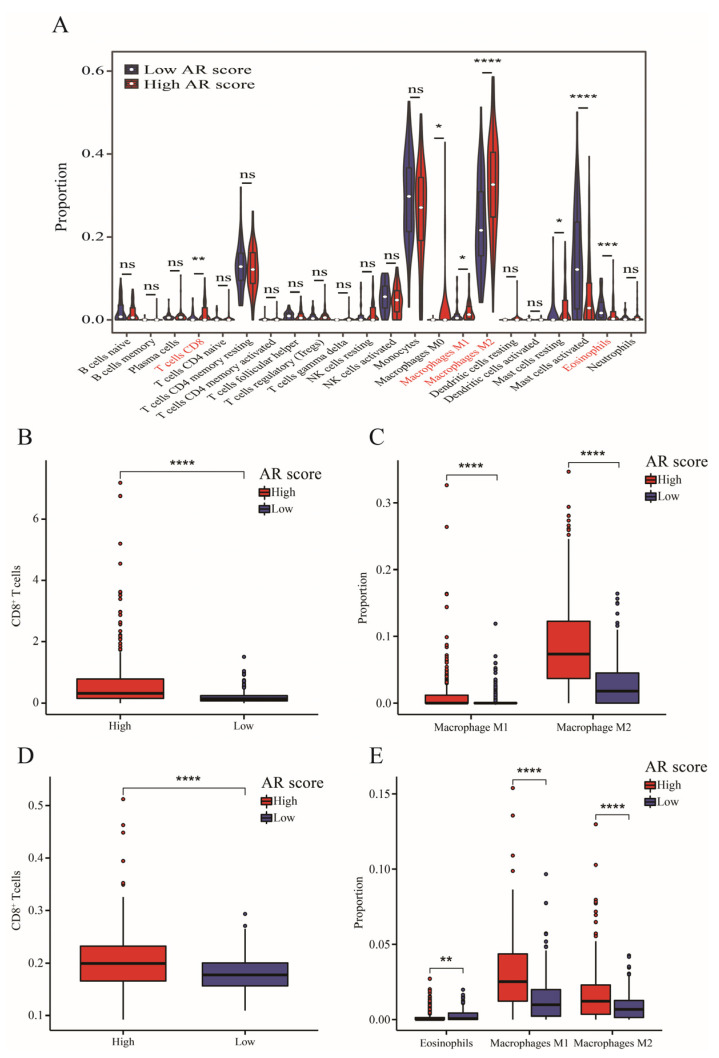
The immune infiltrating analysis. (**A**) The infiltrating level of 22 immune cells between high- and low-AR groups via “CIBERSORT” algorithm. (**B**–**E**) Validation of immune infiltrating results through “MCP-counter”, “quanTIseq”, “TIMER” and “xCell” algorithms. Data in (**A**–**E**) were analyzed by Wilcoxon test; ns, no significance; * *p*  <  0.05, ** *p*  <  0.01, *** *p*  <  0.001 and **** *p* < 0.0001.

**Figure 6 ijms-23-11971-f006:**
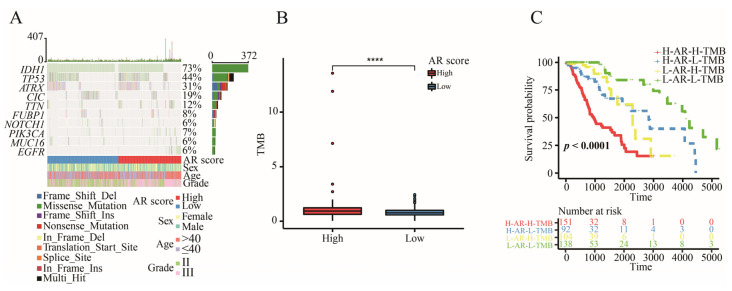
The somatic mutation analysis. (**A**) The waterfall diagram of top 10 driver genes. (**B**) The level of TMB between high- and low-AR groups. (**C**) Kaplan–Meier survival analysis of OS for LGGs stratified by both TMB and AR score. Data in (**B**) was analyzed by Wilcoxon test; **** *p* < 0.0001.

**Figure 7 ijms-23-11971-f007:**
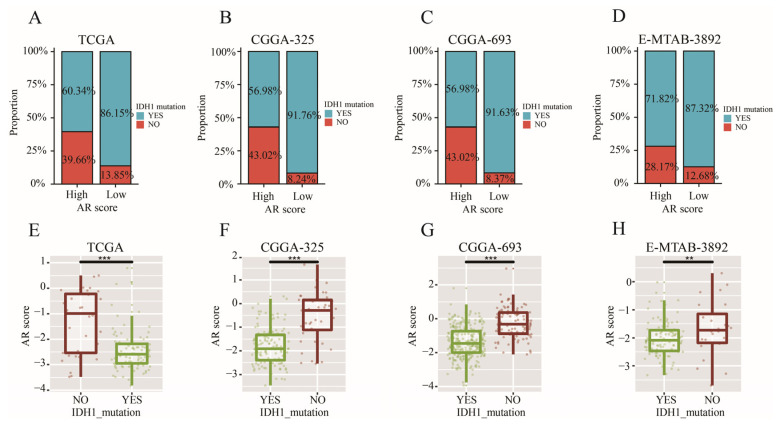
The relationship between AR-related signature and IDH1 mutation. (**A**–**D**) The ratio of IDH1 mutation between high- and low-AR groups in TCGA, CGGA-325, CGGA-693, and E-MATB-3892, respectively. (**E**–**H**) The comparisons of AR scores between mutated IDH1 and wild-type IDH1 in TCGA, CGGA-325, CGGA-693, and E-MATB-3892, respectively. Data in (**E**–**H**) were analyzed by Wilcoxon test; ** *p* < 0.01, and *** *p* < 0.001.

**Figure 8 ijms-23-11971-f008:**
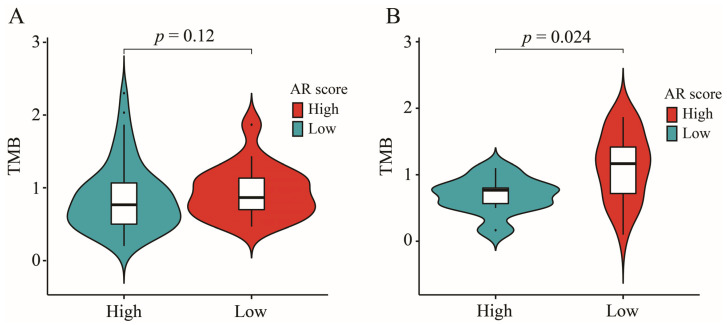
The role of IDH1 mutation on relationship between AR-related signature and TMB. (**A**) The comparisons of TMB between high- and low-AR patients with IDH1 mutation. (**B**) The comparisons of TMB between high- and low-AR patients with wild-type IDH1.

**Figure 9 ijms-23-11971-f009:**
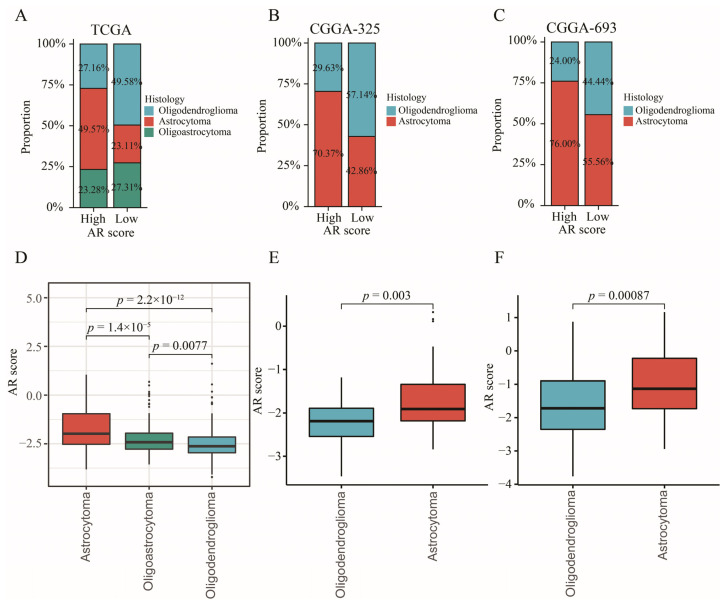
The relationship between AR-related signature and histology. (**A**–**C**) The proportion of histological distribution between high- and low-AR groups in TCGA, CGGA-325, and CGGA-693, respectively. (**D**–**F**) The comparisons of AR scores in different histological subtypes in TCGA, CGGA-325, and CGGA-693, respectively.

**Figure 10 ijms-23-11971-f010:**
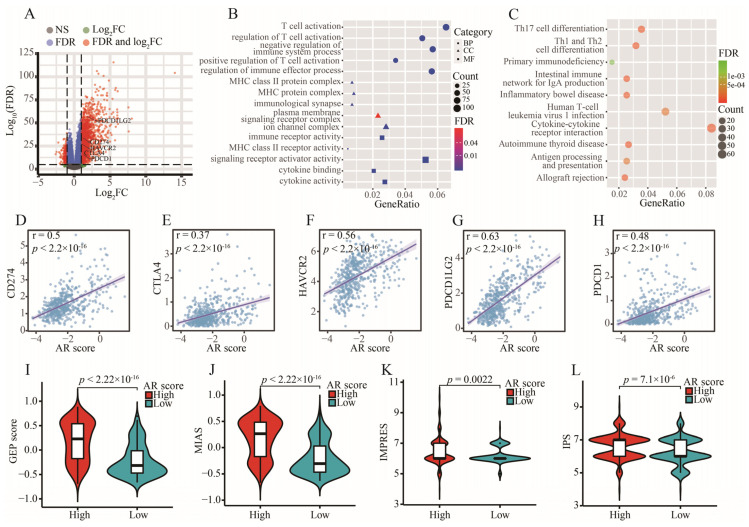
Prediction of immunotherapy using our signature. (**A**) The volcano plot of DEGs between high- and low-AR groups. (**B**,**C**) The GO and KEGG analyses of these DEGs between high- and low-AR groups. (**D**–**H**) The correlation of five ICs (CD274, CTLA4, HAVCR2, PDCD1LG2 and PDCD1) with AR scores. (**I**–**L**) The expression of four ICI-related indicators (GEP, MIAS, IMPRES and IPS) between high- and low-AR groups.

**Figure 11 ijms-23-11971-f011:**
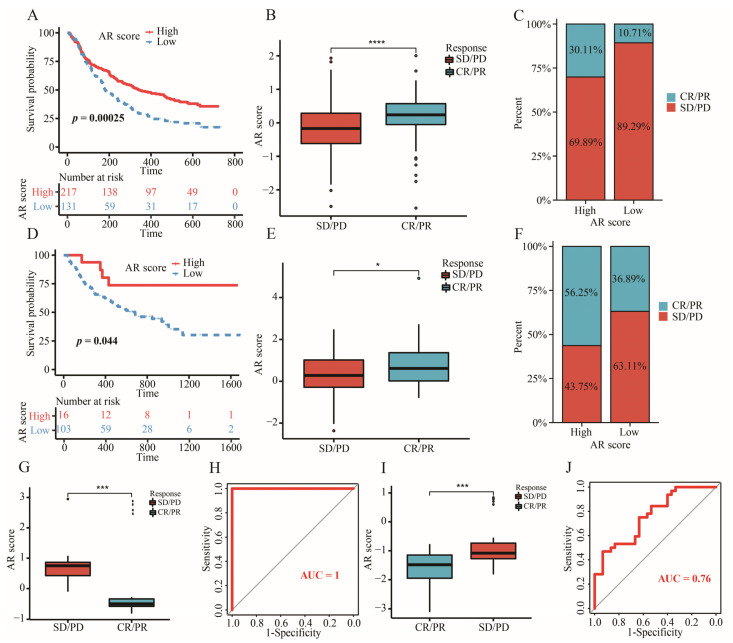
The significance of our signature in various external immunotherapy cohorts. (**A**) Kaplan–Meier survival analysis of OS between high- and low-AR groups in IMvigor210 cohort. (**B**) The comparison of AR scores between CR/PR and SD/PD groups in IMvigor210 cohort. (**C**) The clinical response rate between high- and low-AR groups in IMvigor210 cohort. (**D**) Kaplan–Meier survival analysis of OS between high- and low-AR groups in Liu’s cohort. (**E**) The comparison of AR scores between CR/PR and SD/PD groups in Liu’s cohort. (**F**) The clinical response rate between high- and low-AR groups in Liu’s cohort. (**G**) The comparison of AR scores between CR/PR and SD/PD groups in GSE79671. (**H**) The AUC of predicting patients’ response to bevacizumab treatment in GSE79671. (**I**) The comparison of AR scores between CR/PR and SD/PD groups in GSE148476. (**J**) The AUC of predicting patients’ response to ICIs in GSE148476. Data in (**B**,**E**,**G**,**I**) were analyzed by Wilcoxon test; * *p* < 0.05, *** *p* < 0.001 and **** *p* < 0.0001.

**Figure 12 ijms-23-11971-f012:**
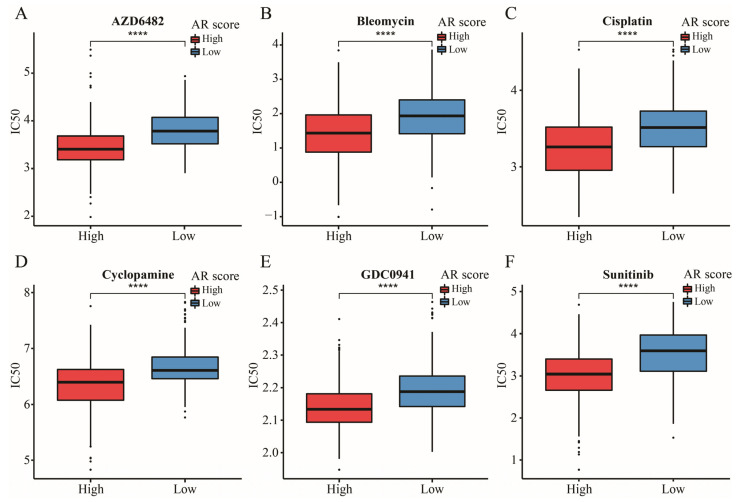
The IC50 of six classic chemotherapeutic drugs between high- and low-AR groups. Data in (**A**–**F**) were analyzed by Wilcoxon test; **** *p* < 0.0001.

**Table 1 ijms-23-11971-t001:** The characteristic of data in this study.

Datasets	Sources	Data Types	Samples
TCGA-LGG	TCGA	gene expression RNAseq	509
CGGA-325	CGGA	mRNA sequencing	172
CGGA-693	CGGA	mRNA sequencing	420
Rembrandt	CGGA	mRNA microarray data	121
E-MTAB-2768	ArrayExpress	array assay	63
E-MTAB-3892	ArrayExpress	array assay	142
Total	/	/	1427

## Data Availability

The corresponding data and results were generated by TCGA (https://xenabrowser.net/, accessed on 1 May 2022), CGGA (http://www.cgga.org.cn/, accessed on 1 May 2022), ArrayExpress (https://www.ebi.ac.uk/arrayexpress/, accessed on 1 May 2022), Ensembl (http://asia.ensembl.org/, accessed on 1 May 2022), GSEA (https://www.gsea-msigdb.org/, accessed on 3 May 2022), HPA (http://www.proteinatlas.org/, accessed on 10 May 2022), GEPIA (http://gepia.cancer-pku.cn/, accessed on 4 May 2022), CIBERSORT (https://cibersort.stanford.edu/, accessed on 15 May 2022) and GEO (http://www.ncbi.nlm.nih.gov/geo/, accessed on 1 May 2022) databases.

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
