# Peer review of "Development of a Hallmark Pathway-Related Gene Signature Associated with Immune Response for Lower Grade Gliomas"

_ijms, 2022, doi:10.3390/ijms231911971_

Round 1

Reviewer 1 Report

The article Title "Development of a hallmark pathway-related gene signature associated with immune response for lower grade gliomas", authored by Guichuan Lai and colleagues provides evidence on the role of an immune profile associated with lower-greade gliomas.

My Comments and Suggestions for Authors are following:

Major concers

1) the study use state-of-the-art bioinformatic tools to predict and suggest a signature, based on RNA-seq data. However, all the infered calculation do not support a direct interaction, unless the authors expreimentaly validate these signatures in an independent patient cohort. As such, I would suggest the authors to tone down the narrative to a level where the conclusions are supported by the results. In this case, the presented data support the existence of a possible immune-signature associated footprint that might be associated with immune response in lower grade gliomas. This change should be mirrored throughout the text.

2) It is not clear which pipeline the authors follow. There should be sufficient information to replicate the calculations made by the authors, in order to validate their claims. I suggest that a detailed flowchart explaining each step of the methodology should be added in the text, preferably as a supplementary file.

minor concerns

1) The authors should consider renumbering their figures. Figure 2 appears first and Figure 1 last. Each Figure should be added in sequential order of appearance to avoid confusion. 

2) there are few errors throughout the text. The authors should consider revising.

Reviewer 2 Report

The authors showed the pathway-related gene signature for lower grade gliomas (LGGs) in association with immune response. The authors found that an allograft rejection (AR)-related pathway is a crucial signaling pathway and showed IC50 values of the six classic chemotherapeutic drugs according to this signature. They claim that this signature can be regarded as an underlying biomarker in predicting prognosis for LGGs. Though this paper is well written and shows deeper insights into the biology of LGGs, some criticisms have arisen.

1.     The authors should start with figure 1.

2.     Although written in Materials and Methods, the authors should describe “these patients receiving radiotherapy” in brief. (line 102)

3.     The authors investigated the expression of five AR-related genes and those were higher in tumor tissue than in corresponding normal tissue. The authors should show and describe the histology of AR-high and low groups. Can they be detected by HE stain?

4.     The authors should correct the aspect ratio of Figure 5A.

5.     Since IDH1 mutation is necessary for molecular and genetic classification of LGGS, the authors should describe the connection between AR-related signature and IDH1 mutation more in detail.

Round 2

Reviewer 1 Report

The authors have succesfully addressed my comments